elLIFE

# Resolving coiled shapes reveals new reorientation behaviors in *C. elegans*

Onno D Broekmans[1], Jarlath B Rodgers[2,3], William S Ryu[2,3,4], Greg J Stephens[1,5*]

[1]Department of Physics and Astronomy, VU University Amsterdam, Amsterdam, The Netherlands; [2]Donnelly Center, University of Toronto, Toronto, Canada; [3]Department of Cell and Systems Biology, University of Toronto, Toronto, Canada; [4]Department of Physics, University of Toronto, Toronto, Canada; [5]OIST Graduate University, Onna, Okinawa, Japan

**Abstract** We exploit the reduced space of *C. elegans* postures to develop a novel tracking algorithm which captures both simple shapes and also self-occluding coils, an important, yet unexplored, component of 2D worm behavior. We apply our algorithm to show that visually complex, coiled sequences are a superposition of two simpler patterns: the body wave dynamics and a head-curvature pulse. We demonstrate the precise $\Omega$-turn dynamics of an escape response and uncover a surprising new dichotomy in spontaneous, large-amplitude coils; deep reorientations occur not only through classical $\Omega$-shaped postures but also through larger postural excitations which we label here as $\delta$-turns. We find that omega and delta turns occur independently, suggesting a distinct triggering mechanism, and are the serpentine analog of a random left-right step. Finally, we show that omega and delta turns occur with approximately equal rates and adapt to food-free conditions on a similar timescale, a simple strategy to avoid navigational bias.

*For correspondence: g.j. stephens@vu.nl

Competing interests: The authors declare that no competing interests exist.

## Introduction

Much of our fascination with the living world, from molecular motors to the dynamics of entire societies, is with emergence — where the whole is surprisingly different than the sum of its parts (see, e. g., [*Laughlin, 2014*]). Yet, the existence of such collective organization also suggests that living systems, despite their enormous potential complexity, often inhabit only a much smaller region of their potential 'phase space', and evidence for this lower-dimensional behavior is ubiquitous. For example, the motor control system produces movements that are far less complex than what the musculoskeletal system allows (*d'Avella et al., 2003*) and this hints at the presence of an organizational principle. In a typical daily movement like walking, the central nervous system is thought to produce the full walking gait by combining low-level 'locomotory modules', some of which appear to be universal among species (*Dominici et al., 2011*). Similarly, the dynamics in brain networks are organized in low-dimensional activity patterns (*Tkačik et al., 2014*; *Gao and Ganguli, 2015*) and these patterns — not individual neurons — might be the carriers of information and computation (*Hopfield, 1982*; *Yoon et al., 2013*).

The emergent dynamics of behavior, how animals move and interact, is particularly important as the ultimate function of the system (*Tinbergen, 1963*) and the scale on which evolution naturally applies. Yet, our quantitative understanding of behavior is substantially less advanced than the microscopic processes from which it is produced, even as recent efforts have expanded this frontier (*Mirat et al., 2013*; *Berman et al., 2014*; *Cavagna and Giardina, 2014*). How do we analyze high-resolution behavioral dynamics and what does this reveal about an animal's movement strategy? How do we build effective models on the behavioral level where a 'bottom-up' approach is

**eLife digest** We all instinctively recognize behavior: it's what organisms *do*, whether they are single cells searching for food, or birds singing to mark their territory. If we want to *understand* behavior, however, we have to be able to characterize such actions as precisely and completely as their underlying molecular and cellular mechanisms.

For the millimeter-sized roundworm *C. elegans*, video tracking and analysis has produced a compact characterization of naturally occurring worm postures. Simply put: every body posture of the worm is a different mix of four fundamental postures called 'eigenworms'. The worm's snake-like motion is then a series of combinations of these projections, which can be analyzed to provide an automatic and measureable read-out of the worm's behavior.

There is, however, an important caveat: when the worm makes a 'loop', and crosses over itself, such posture analysis is inapplicable. That is unfortunate: some of the worm's most interesting behavior involves looping. One example is the "omega turn", named after the shape of the Greek letter Ω. This sharp turn is used by the worm to steer away from harm, and more generally to abruptly reorient during the search for food and for mates.

Broekmans et al. have now created an algorithm, based on eigenworms, which can analyze worm images that encompass both looped and normal shapes. The result is a complete 'behavioral microscope' that shows how *C. elegans* moves in 2D. Focusing this microscope in particular on the omega turn, Broekmans et al. found that such turns are not, as has been previously described, a single behavior. Instead, they are two separate behaviors that represent the worm's equivalent of a left-right step.

Together with previous posture analysis the work presented by Broekmans et al. allows for the full and precise measurement of the body shapes of *C. elegans* in 2D. This, combined with remarkable recent progress in global brain and gene expression imaging, should help to uncover new mechanisms that ultimately produce and control a worm's behavior.

daunting? How do we connect analysis on the organism-scale to the properties of molecules, cells and circuits? We approach these questions through the postural movements of the nematode *C. elegans*.

In *C. elegans*, the 2D space of body postures can be captured precisely and is also low-dimensional (*Stephens et al., 2008*) so that the worm's motor behavior is faithfully encoded as a time series of only four 'eigenworm' variables. These shape projections are collective coordinates in the space of natural worm postures and provide a notable reduction in complexity. However, an important limitation of previous work is the inability to deduce the geometry of self-occluding body shapes. Such coiled body postures occur during 'omega turns' (a maneuver during which the worm's body briefly resembles the Greek letter Ω [*Croll, 1975*]) and are a general and important feature of the worm's behavioral repertoire, ranging from foraging (*Stephens et al., 2010*; *Salvador et al., 2014*), and chemotaxis (*Pierce-Shimomura et al., 1999*), to escape from noxious stimuli (*Mohammadi et al., 2013*). For example, during escape behaviors worms use coiled shapes to reorient by 180° and the benefit seems obvious: it steers the worm back to safety. But how does a 'blind' organism achieve this result without any visual reference to the outside world? While some of the neural and molecular mechanisms driving omega turns have been uncovered (*Gray et al., 2005*; *Donnelly et al., 2013*) and there has been previous work on crossed shapes (*Huang et al., 2006*; *Wang et al., 2009*; *Roussel et al., 2014*; *Nagy et al., 2015*), a quantitative analysis of such self-occluded posture dynamics is lacking.

Here, we exploit low-dimensionality to develop a novel and conceptually simple posture tracking algorithm able to unravel the worm's self-occluding body shapes. We apply our approach to analyze coiled shapes during two important behavioral conditions: the escape response induced by a brief heat shock to the head, and spontaneous turns while foraging on a featureless agar plate. We find that, in general, complex deep turn sequences can be viewed as a simpler superposition of body wave phase dynamics with a bimodal head swing followed by a unimodal curvature pulse. In the escape response we show that, while turning accounts for much of the ~180° reorientation, the full

distribution of reorientation angles is shaped by significant contributions from the reversal, turn and post-turn behaviors, a result consistent with the presence and action of the monoamine tyramine during the entire response (*Donnelly et al., 2013*). In natural crawling, the peak amplitudes of the curvature pulse reveal two distinct coiling behaviors — the classical omega turn accomplishing large ventral-side reorientations, and a previously uncharacterized 'delta' turn which produces dorsal reorientations by overturning through the ventral side. The omega and delta turns occur independently in time, suggesting a separate triggering process, but have similar rates, as expected if they contribute little overall bias in the trajectories.

## Results

### Tracking posture using low-dimensional worm shapes

Previously, we analyzed movies of *C. elegans* freely crawling on an agar plate (*Figure 1A*) (*Stephens et al., 2008*). For each movie frame, we identified the body of the worm, and applied a thinning algorithm to find the centerline. The worm's 2D body posture was characterized as a 100-dimensional vector of tangent angles along this centerline (*Figure 1B–C*). Principal Component Analysis revealed that more than 95% of the variance in naturally-occurring body postures was captured by just four eigenvectors of the posture covariance matrix (*Figure 1D*). As a result, any worm posture can be decomposed as a linear combination of these 'eigenworms' (*Figure 1E*). Worm behavior then becomes a smooth, low-dimensional trajectory through posture space (*Figure 1F*). As an example, forward and backward crawling appear as approximately circular trajectories in the $(a_1, a_2)$ plane, and correspond to limit-cycle attractors. However, for coiled shapes such as shown in *Figure 1H*, the thinning algorithm does not produce a faithful reconstruction of the worm's actual posture (*Figure 1G*).

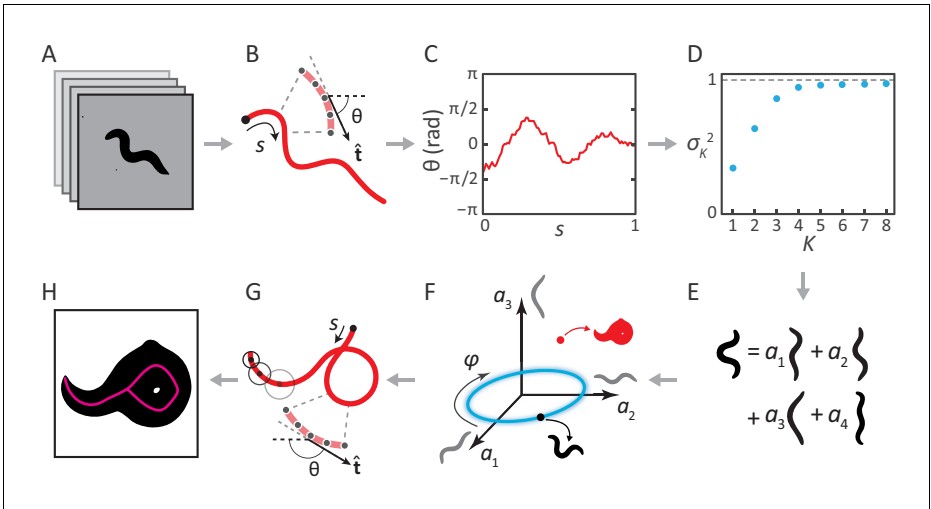

**Figure 1.** Inverting posture analysis to generate worm images. (**A–E**) We previously showed that the space of *C. elegans* body postures is low-dimensional. (**A**) For a set of images of a freely moving worm, (**B**) we find the centerline of the body using image thinning (black point indicates the head). (**C**) At equidistant points along the centerline, we measure the direction $\theta(s)$ of the tangent $\hat{t}$. After subtracting $\langle\theta\rangle$, this gives a description of the worm's shape that is intrinsic to the worm itself. (**D**) Principal Component Analysis reveals that only four eigenvectors of the shape covariance matrix are needed to account for $\sim 95\%$ of the variance in $\theta(s)$. (**E**) Hence, any body shape can be decomposed as a linear combination of postural 'eigenworms'. (**F**) Alternatively, we can think of any body posture as a point in a low-dimensional 'posture space', spanned by the eigenworms (gray). Forward crawling is then represented by clockwise progression along a circular trajectory in the $(a_1, a_2)$ plane (blue oval, body wave phase angle $\varphi$). (**G**) For any point in this space, we can easily calculate the shape of the backbone. A series of filled circles with radii representing the worm's thickness, are used to draw an image of the worm's body (**H**), inverting the original postural analysis to generate an image. For self-overlapping shapes, image thinning (*H, magenta*) does not produce an accurate reconstruction of the posture (*G, red*).

The above procedure can also be implemented in reverse to *generate* worm images. For any point $p$ in posture space (*Figure 1F*), we can reconstruct the shape of the backbone (*Figure 1G*). Knowing the thickness of the worm at each point along the body (which we estimate by averaging over many worm images), we are able to draw a reconstructed body image (*Figure 1H*; see Materials and methods). We then track the posture by finding, for each movie frame, the point in posture space (and thus the correct centerline) for which the reconstituted worm image is the most similar to the original image. This approach works for all worm postures — in contrast to image thinning, which fails for self-overlapping shapes (*Figure 1H*).

Our 'inverse' tracking algorithm consists of three basic elements. (i) An image error function $f_{err}$ quantifies how well a reconstituted worm image $\tilde{\mathbf{W}}(p)$ matches the movie frame $\mathbf{W}$ (*Figure 2A*); (ii) an efficient optimization scheme to search for a global error minimum over all possible postures, and; (iii) a method to resolve ambiguity, as different self-occluding body shapes can give rise to the same image. We measure image similarity using two specific shape metrics (*Yang et al., 2008*): outline shape, and coarse-grained pixel density (*Figure 2A*). By mapping this error function onto posture space: $f_{err}(p) = f_{err}(\mathbf{W}, \tilde{\mathbf{W}}(p))$, we create a fitness landscape, in which the position of the global minimum corresponds to the tracking solution. We find this minimum using a pattern search algorithm (a form of direct search [*Kolda et al., 2003*]). To resolve ambiguity, we retain multiple minima for each frame, until a final step which minimizes total sequence error. We sketch this process for a single mode in *Figure 2C*.

## Tracking reproduces both simple and self-occluding worm shapes with small errors

Tracking results for a typical movie that includes complex, self-occluding shapes are shown in *Figure 2D* (see also *Videos 1* and *2*). In the gray rows at the top are the original movie frames; the reconstituted images from our inverse algorithm are below. While some minor inaccuracies are visible by eye, the overall result is remarkably similar. To quantify posture tracking accuracy, we first compared the results of our algorithm to image thinning, which allows for verification based on a large dataset. We used image thinning to construct a 100-dimensional vector of tangent angles $\boldsymbol{\theta}$, defined the tracking 'error' as $\delta\theta = \left\| \boldsymbol{\theta}_{inv} - \boldsymbol{\theta}_{thinning} \right\|$, and we plot the distribution of these errors in *Figure 2E* (magenta). We also show the discrepancy in $\boldsymbol{\theta}$ that results from dimensionality reduction to the postural eigenmodes (black). Additionally, we show Euclidean distances between tangent angle vectors of consecutive frames in a 16 Hz movie, representing limited time resolution (gray). For this dataset of non-crossed frames, our algorithm provides excellent performance, with tracking errors bounded by time resolution and dimensionality reduction. Even for deviations in the tail of the distribution ($\delta\theta = 3\,\mathrm{rad}$), backbones from the thinning and the 'inverse' algorithm are quite similar (inset, gray backbones).

A more relevant quantity for low-dimensional trajectories is the mode discrepancy $\delta a_i = \left\| a_i^{inv} - a_i^{thinning} \right\|$ which is negligible for simple shapes, as shown in *Figure 2F* (yellow). Finally, we created a dataset of self-overlapping body shapes for which backbones were manually drawn. In *Figure 2F* (blue), we show that, for the majority of crossed frames, the mode error is less than 10% of the total range of naturally occurring mode values. As a visual reference, the reconstituted worm shapes corresponding to mode errors of $\delta a_i = 1$ are shown in gray: these are noticeably flat.

## Coiled dynamics in the escape response reveal precise reorientations and the superposition of the body wave and a head-curvature pulse

We first applied our postural tracking algorithm to quantify the full shape dynamics of the *C. elegans* 'escape response'. This is a stereotyped behavioral sequence, consisting of a pause, a reversal and an $\Omega$-turn, that quickly moves the worm away from a threatening stimulus. Featuring only relatively simple coiled shapes, the escape response provided a useful test of our algorithm. While recent work has connected the escape response with genetic, molecular, and neural mechanisms (*Donnelly et al., 2013*), the behavior itself has been described only qualitatively. Here, we elicited an escape response by using an infrared laser pulse administered to the head of the worm, which raised the temperature by ~0.5℃. 10 s of pre-stimulus behavior and 20 s of post-stimulus behavior

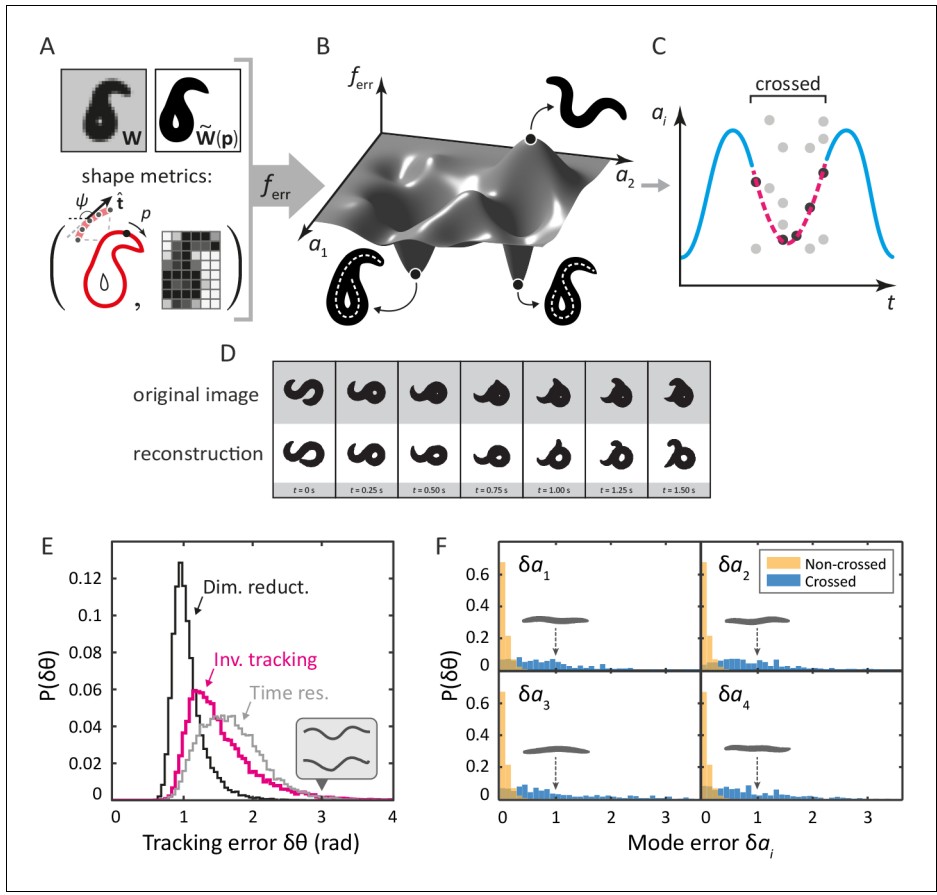

**Figure 2.** Tracking coiled shapes by searching for image matches in posture space. Top: tracking algorithm sequence. (A) For each movie frame $\mathbf{W}$ and reconstituted worm image $\tilde{\mathbf{W}}(\boldsymbol{p})$ for posture $\boldsymbol{p}$, we apply two metrics, one based on the shape of the boundary (left), and one based on a coarse-grained pixel density matrix (right). (B) An error function $f_{\mathrm{err}}$ based on these two shape metrics generates a fitness landscape (schematically shown). The position of the global minimum of $f_{\mathrm{err}}$ corresponds to the tracking solution; if a frame is ambiguous, multiple minima may be present. (C) For non-crossed body postures, a simple image thinning algorithm suffices to obtain time series of the modes $a_i$ (blue line, schematically shown). For crossed frames, we use the procedure outlined in A–B. Due to the inherent ambiguity of such images, multiple solutions are generally found for each frame (light gray points). Using the filtering algorithm described in the Materials and methods, we identify the correct solutions (dark gray points). The resulting smooth trajectory (magenta, dotted line) forms the full tracking solution. (D) Sample tracking results (bottom, white background), contrasted with original images (top, gray background), for a turning sequence. Bottom: the inverse algorithm accurately tracks both simple and coiled worm shapes with small error. (E) Histogram of tracking errors for non–self-overlapping worm shapes, quantified as the Euclidean distance $\delta\theta$ between the tangent angle vector $\boldsymbol{\theta}$ from our algorithm, and $\boldsymbol{\theta}$ found by image thinning (magenta). For scale, the error due to dimensionality reduction to five postural eigenmodes is shown in black. We also show the Euclidean distance between $\boldsymbol{\theta}$ in consecutive frames, representing the confidence in $\boldsymbol{\theta}$ due to the finite time resolution of the movie (gray). Even for an extreme value of $\delta\theta = 3\,\mathrm{rad}$ (gray arrow), backbones from the 'classic' algorithm (top) and our algorithm (bottom) are nearly indistinguishable by eye (inset). (F) Tracking error in eigenmode values for the first four modes. For uncrossed worm shapes (yellow/light), our algorithm shows negligible tracking errors. For a smaller set of crossed frames, we compare to a manually found solution (blue/dark). For scale, we show reconstituted images for worms with a single nonzero mode value of $a_i = 1$; these 'error worms' are essentially flat.

The following figure supplement is available for figure 2:

**Figure supplement 1.** The eigenworms $\{e_{k=1...4}\}$ derived from the fully-tracked data show only minor changes compared to those computed without crossings.

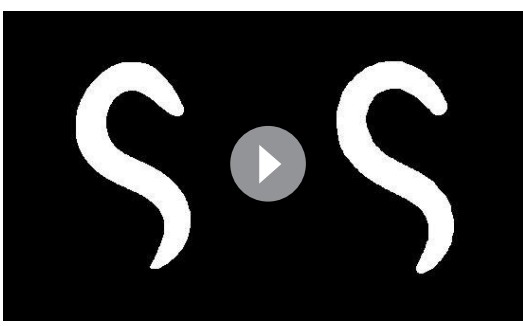

**Video 1.** Tracking results for the escape response. Left images are data, while on the right, there are reconstructed images from our tracking algorithm.

were recorded at $20\,\mathrm{Hz}$. Each worm was only assayed once, to prevent adaptation. In total, $N = 92$ worms were recorded, of which $N = 91$ successful trackings were used in the final analysis.

A schematic of the response is shown in *Figure 3A*, with the associated postural mode dynamics in *Figure 3B,C*. During normal, forward locomotion (i in *Figure 3A*, $t<10\,\mathrm{s}$ in *Figure 3C*), the worm crawls by propagating a sine-like wave through its body. This is reflected as a pair of phase-locked sinusoidal oscillations in $a_1$ and $a_2$ and we define the body wave phase angle $\varphi = -\arctan(a_2/a_1)$, where the minus sign ensures that $d\varphi/dt$ is positive during forward crawling. When the worm is stimulated by the infrared pulse (ii in *Figure 3A*, pink line in *Figure 3C* at $t$ = 10 s), it immediately backs up (iii), seen as a decrease in $\varphi$. The end of this reversal and the beginning of the $\Omega$-turn is marked by a head-swing, visible as a bimodal pulse in $a_4$. The $\Omega$-turn itself (iv) occurs as a large, unimodal pulse in $a_3$, and propagates head-to-tail. This implies another switch of the direction of the body wave, and hence a return to increasing $\varphi$. Finally, as the turn is finished, the worm resumes forward crawling (v). The mode dynamics outlined above illustrate that the complexity of the escape sequence can be seen as a superposition of two simpler patterns: the body wave phase dynamics in $(a_1, a_2)$, and the head-curvature dynamics of $(a_3, a_4)$. An animation of these mode dynamics is available as *Video 3*.

A notable feature of the escape response is how closely the worm controls its reorientation. Our tracking algorithm also makes it possible to track the overall orientation continuously, across the different phases of the escape response. In *Figure 3D–E*, we calculate how much each of the three response segments reorients the worm. The distribution of reorientations for the full escape response is largely similar to the distribution during the omega turn, but includes contributions from the reversal and post-turn segments. In the trial-averaged reorientation *Figure 3E*, we find $\langle \Delta\theta \rangle = -0.89\pi \pm 0.05\pi\,\mathrm{rad}$ for the full response. The omega turn itself results in $\langle \Delta\theta \rangle = -0.90\pi \pm 0.04\pi\,\mathrm{rad}$, while pre- and post-omega phases show smaller but significant contributions, $\langle \Delta\theta \rangle = 0.13\pi \pm 0.03\pi\,\mathrm{rad}$ and $\langle \Delta\theta \rangle = -0.12\pi \pm 0.03\pi\,\mathrm{rad}$, respectively (errors are calculated using bootstrap across trials and are equivalent to standard errors of the mean). In *Figure 3D*, the interval $(0, -\pi)$ corresponds to a final ventral-side reorientation, and $(-\pi, -2\pi)$ to a final dorsal-side reorientation. The small number of reorientations between $(0, \pi)$ are also final dorsal-side reorientations but are achieved using a shallow dorsal bend, not an omega turn, and excluding these worms results in a total mean reorientation angle $\langle \Delta\theta \rangle = -0.97\pi \pm 0.04\pi\,\mathrm{rad}$.

Remarkably, the mean reorientation in the reversal and post-turn segments precisely cancel, suggesting a correction mechanism at the level of the average response so that the mean overall reorientation is entirely determined by the omega-turn. No such precision is apparent in the variance, where we find $\delta\theta^2 = 0.69\pi \pm 0.16\pi\,\mathrm{rad}^2$ for the full response compared to the smaller $\delta\theta^2 = 0.45\pi \pm 0.16\pi\,\mathrm{rad}^2$ for the turn segment. Thus, while the omega turn is an effective maneuver for turning away from the stimulus, the full response orientation change is broadened by the reversal $\delta\theta^2 = 0.23\pi \pm 0.05\pi\,\mathrm{rad}^2$ and post-omega $\delta\theta^2 = 0.19\pi \pm 0.04\pi\,\mathrm{rad}^2$ behaviors.

These observations allow us to hypothesize a subtle link between the behavior of the worm and the escape response at the neurotransmitter level (*Donnelly et al., 2013*). As the worm enters

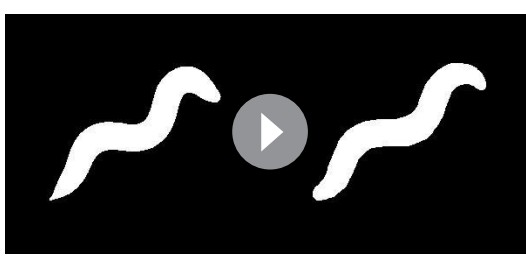

**Video 2.** Tracking results for a complex, spontaneous coil. Left images are data, while on the right, there are reconstructed images from our tracking algorithm.

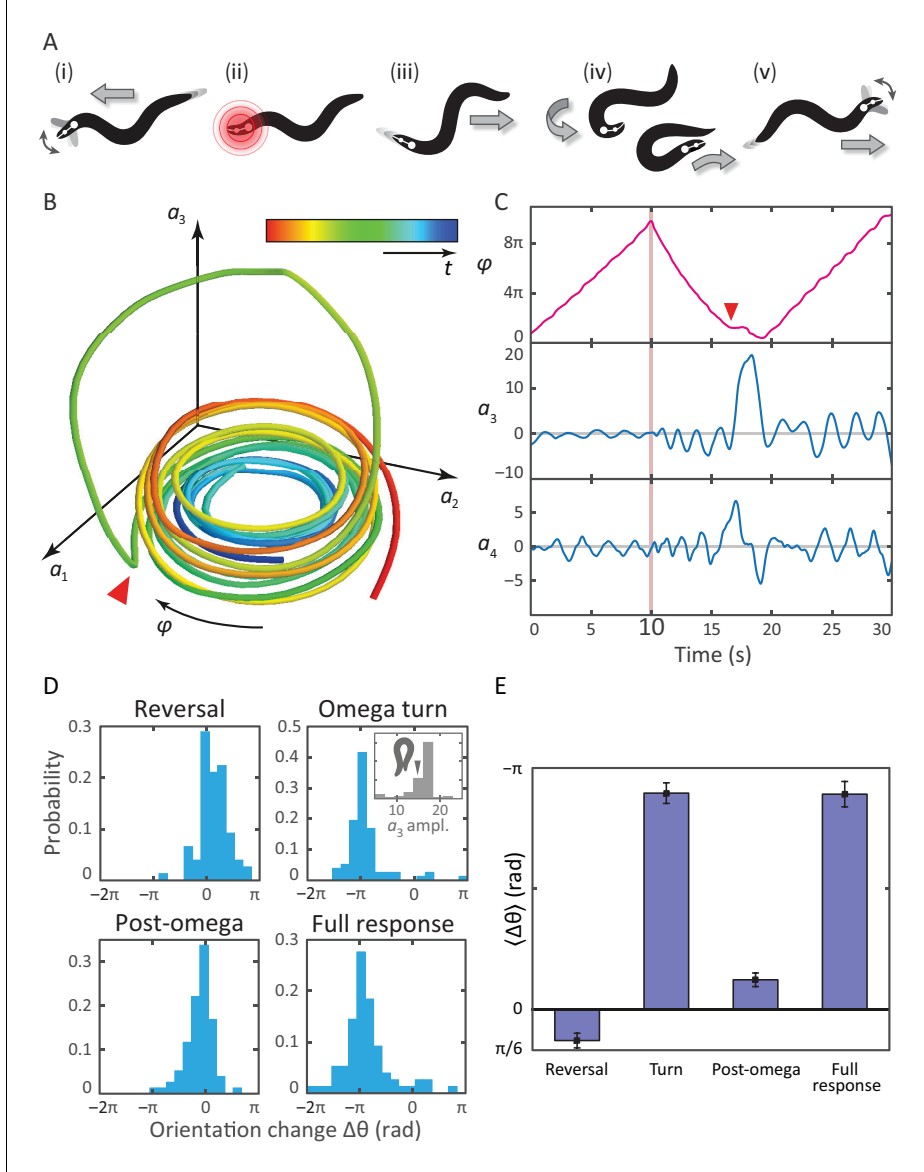

**Figure 3.** Tracking coiled postures and reorientation in the escape response. (**A**) Schematic overview (***Donnelly et al., 2013***) with worm body shapes extracted from tracking data: *i* forward locomotion and exploratory head motions; *ii* infrared laser stimulus; *iii* reversal phase; *iv* omega turn; *v* resumption of forward locomotion in the opposite direction. (**B**) Trajectory through posture space. $\varphi$ indicates direction of increasing body wave phase angle, and color encodes time, with blue for $t = 0$ and red at $t = 30$ s. The worm's reorienting coiling behavior is evident as a large excursion along the third mode, starting at the red arrow. (**C**) The same trajectory as in *B*, in terms of the body wave phase angle $\varphi$ and the postural modes ($a_3, a_4$). The heat shock occurs at $t = 10$ s (pink bar). The omega turn is initiated by a head swing, as seen in $a_4$, followed by a large pulse in $a_3$, and is linked to a 're-reversal', a return to forward movement. (**D**) An important feature of the escape response is the change in the worm's overall orientation, and we apply our algorithm to track this reorientation for each response segment. While turning accounts for much of the reorientation, the full response distribution is shaped by significant contributions from all three segments. In particular, the small but biased reorientations of the reversal and post-turn segments originate in the $a_3$ fluctuations outside the turn (see the time series in *C* and also ***Figure 3—figure supplement 1***). This is consistent with the release and presence of the monoamine tyramine during the entire response. (**E**) The precision of the escape response is evident in the trial-mean reorientation where we find $\langle\Delta\theta\rangle = -0.89\pi \pm 0.05\pi$ rad for the full response and $\langle\Delta\theta\rangle = -0.97\pi \pm 0.04\pi$ rad if we exclude (four) worms that only make small dorsal reorientations. Notably, the mean reorientation in the reversal and post-turn segments closely cancel, suggesting a correction mechanism at the level of the average response. In the inset to
*Figure 3 continued on next page*

*Figure 3 continued*

(**D**, Omega turn), we also show the distribution of $a_3$ amplitudes, which is peaked near coiled shapes in which the worm barely touches.
The following figure supplement is available for figure 3:

**Figure supplement 1.** Bias in the turning mode $a_3$, and resulting reorientation, occurs during all epochs of the escape response.

the reversal phase, release of tyramine sets up an asymmetry in the worm's body, and this appears as a baseline shift in the fluctuations of the third mode (see also *Figure 3—figure supplement 1*) leading to a positive bias in the reorientation, *Figure 3D,E* (reversal). After the turn, lingering effects of the tyramine produce a similar baseline shift, but as the worm is moving forward instead of backward, this now leads to an opposite orientation bias, *Figure 3D,E* (post-omega).

## Coiled dynamics in foraging reveal a surprising dichotomy in large-amplitude turns

To analyze more complex coiled shapes, we applied our posture algorithm to foraging worm behavior on a flat agar plate. Under these conditions, worms navigate using a combination of maneuvers (*Gray et al., 2005*), including short and long reversals, pirouettes and also gradual turns (*Iino and Yoshida, 2009*). We are particularly interested in the pirouettes, as they involve deep coils. Such body bends are primarily encoded in the third postural eigenmode ($a_3$) and, as discussed in the previous section, peaks in $a_3$ are a characteristic feature of omega turns, and have a known role in reorientation of the worm (*Stephens et al., 2010*).

In *Figure 4A*, we show the full distribution of postural mode $a_3$ for all local extrema. Note that the modes have been normalized so that negative $a_3$ amplitudes correspond to dorsal turns; ventral turns have strictly positive amplitudes. A clear asymmetry can be observed so that on top of a symmetric background distribution of shallow turns in both directions, we see, on the ventral side, two distinct additional peaks. Drawing reconstituted worm images for the center values of these two peaks, it is clear that the peak at $a_3 \sim 15$ corresponds to a 'classic' $\Omega$ shape. The second peak, at $a_3 \sim 23$, shows a body shape with a much higher characteristic curvature. In *Figure 4A* (right), we have 'folded' the dorsal side of the distribution over the ventral side, highlighting the ventral asymmetry at high $a_3$ amplitudes. As noted in the figure, we refer to turns in the lower-amplitude peak as *omega turns* and distinguish these from the higher-amplitude *delta (δ) turns* in the second peak. As for the omega turn, the name *delta turn* is chosen to reflect the δ-like shape of the worm during a typical sequence.

Returning to the original tracking movies, the presence of these two classes of turns is clearly visible. In *Figure 4B*, we display movie stills for two example turns: one omega turn, and one delta turn. During the classical omega turn, the worm slides its head along its body, similar to the escape response, ending up with a large, primarily ventral reorientation. A delta turn, on the other hand, is much deeper: the worm completely crosses its head over its body, resulting in a dorsal reorientation by 'over-turning' across the ventral side.

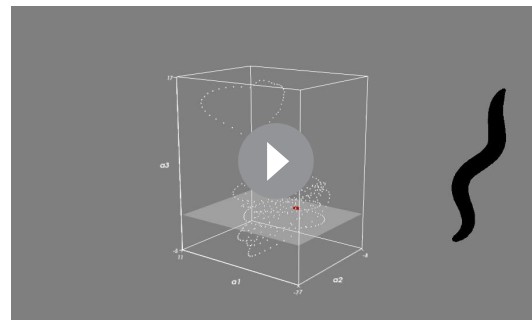

**Video 3.** The dynamics of the escape response in the space of the first three eigenworms. On the right, we show the full body posture, which turns red at the moment of the thermal stimulus. On the left are the dynamics in mode space. The large-amplitude omega turn is visible as a 'figure-8' trajectory. Note that, even during the turn, the body wave is progressing. In general, turning behavior is a superposition of the body wave and curvature dynamics.

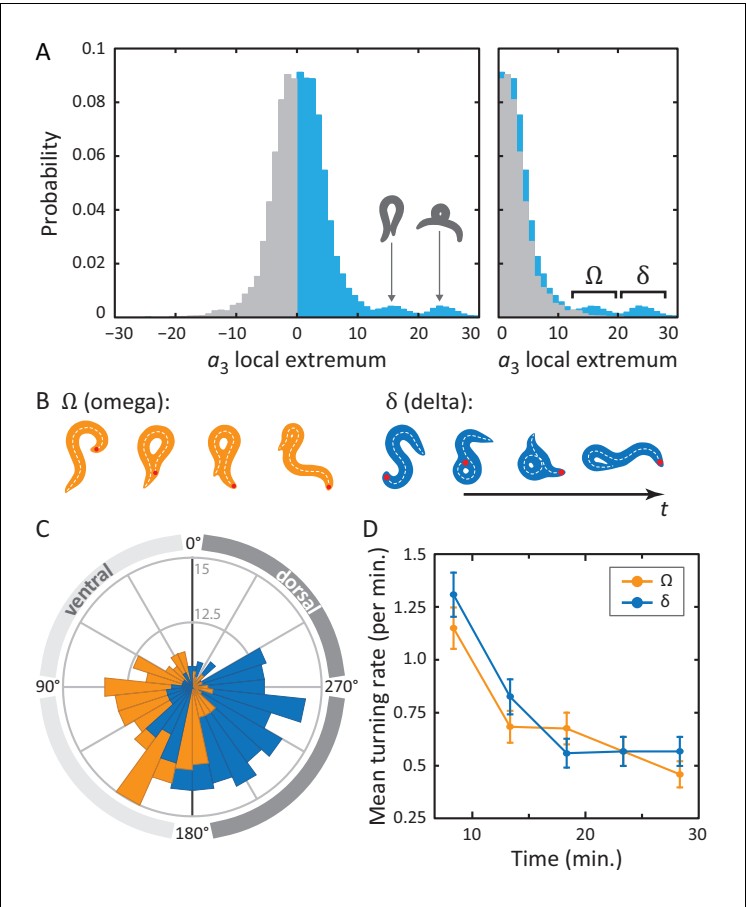

**Figure 4.** Unraveling coiled shapes during foraging reveals two distinct ventrally-biased classes of large-amplitude turns. (**A**) (left) Probability of the amplitude of all local extrema in the time series of the third postural eigenmode $a_3$. Colors represent the sign of the $a_3$ amplitude, and hence the dorsal (gray) or ventral (blue) direction of the resulting turn. (**A**) (right) As previously, with all negative $a_3$ amplitudes now plotted as positive. The peaked excess in the distribution for large ventral bends corresponds to 'classic' $\Omega$ (omega) shapes, and previously undescribed, deeper $\delta$ (delta) turns. Insets in *A* (left) show reconstructed worm shapes for the indicated $a_3$ amplitudes. (**B**) Stills from a movie of a worm making a classical omega turn (left, yellow), and a deep delta turn (right, blue). The head is marked with a red dot; dashed lines indicate postures determined from our inverse tracking algorithm. The dynamics of delta turns are largely similar to omega turns, differing primarily in the amplitude of the bending mode $a_3$, and the overall time to complete the maneuver (see *Figure 4—figure supplement 1*). (**C**) Histogram of orientation change ($\Delta\langle\theta\rangle$) due to ventral omega turns (yellow/light) and ventral delta turns (blue/dark). Ventral reorientations are accomplished through omega turns. To reorient to the dorsal side, however, *C. elegans* employs delta turns, which 'over-turn' through the ventral side. (**D**) Average turning rate during the tracking experiment. Ventral omega and delta turns are temporally independent, suggesting a separate triggering mechanism, but occur with approximately equal rates that adapt similarly with time spent away from food, a simple strategy to avoid any dorsal-ventral navigational bias.

The following figure supplements are available for figure 4:

**Figure supplement 1.** Omega and delta turns follow similar kinematics; while visually quite distinct, the primary difference is the amplitude of the curvature pulse $a_3$.

**Figure supplement 2.** The shifted mutual information between delta turn and omega turn time series.

**Figure supplement 3.** (left) Location of one of the 12 tracked worms over the course of a 35-min tracking experiment (off-food), starting at (0, 0) (black arrow).

## Delta and omega turns are the serpentine analog of a left-right step and occur independently in a navigational strategy

In postural dynamics, $\delta$- and $\Omega$-turns differ primarily in their $a_3$ pulse amplitude; their turn kinematics are otherwise very similar (*Figure 4—figure supplement 1*). However, when turns do occur, they result in a dramatically different change of overall orientation. As in the escape response, we use our algorithm to track the worm's overall body reorientation, and in *Figure 4C*, we show how the worm reorients using both omega (orange) and delta (blue) turns. Simply put, omega turns reorient the worm by large, ventral angles, while delta turns reorient the worm dorsally by 'over-turning' through the ventral side. The difference in reorientation angle may provide a hint as to why these two behaviors exist. Earlier, we saw that the neural mechanisms that produce the escape-response omega turn, are fundamentally asymmetric, producing only ventral turns (through disinhibition of the VD motor neurons) (*Donnelly et al., 2013*). If the worm uses the same neural infrastructure during free crawling, this would only ever allow it to reorient itself towards its ventral side. Lacking a dorsal 'copy' of the same neural infrastructure, the worm could instead hyper-activate the existing infrastructure to produce ventral 'over-turning'. These 'over-turns' are what we call delta turns, and enable the worm to also reorient towards its dorsal side. We also find that delta and omega turns occur seemingly independently; the mutual information between time-binned, time-shifted series for both turning event time series has a maximum of less than a few percent (see Materials and methods and *Figure 4—figure supplements 2* and *3*). On the other hand, evidence that the turns can be jointly controlled is shown in *Figure 4D*. Here, we plot the frequency of turning events over the course of the experiment. As the worm searches for food in a larger area, the turn frequency decreases significantly — a well-known phenomenon (*Gray et al., 2005*; *de Bono and Villu Maricq, 2005*; *Srivastava et al., 2009*) — and both omega and delta turns show similar frequencies and adaptation.

## Discussion

The ability to track self-overlapping shapes of *C. elegans* together with the eigenworm projection of postures, provides a complete and quantitative accounting of the worm's locomotory behavior in 2D. Among living systems with a nervous system, such an exact behavioral description is unique, and is likely to be especially important as new techniques emerge for the simultaneous imaging of a substantial fraction of the worm's neurons during free behavior (*Nguyen et al., 2016*; *Venkatachalam et al., 2016*). Our posture tracking algorithm itself is conceptually simple and relies on an optimized image search within the low-dimensional space of worm shapes. Indeed, while the identification of low-dimensionality occupies an important role in quantitative approaches to living systems (see e.g. *Machta et al., 2013*; *Daniels and Nemenman, 2015*; *Ganguli and Sompolinsky, 2012*), here we have leveraged low-dimensionality to elucidate important and previously unknown aspects of *C. elegans* coils. Interestingly, we were able to apply the characterization of body postures developed previously for non–self-overlapping body shapes (*Stephens et al., 2008*), to capture shapes that *do* self-overlap; even the simpler eigenworm space allows for substantial postural diversity.

We applied our tracking algorithm to two important behaviors: an evoked escape response; and the deep, spontaneous turns that occur during foraging. Viewing the coiled turn as a trajectory through the low-dimensional posture space, a simple model emerges: a superposition of the body wave (a circular trajectory in posture space corresponding to simple forward and backward crawling), and coupled pulses along the third and fourth mode (corresponding to the deep coil and a preceding head oscillation). This model is consistent with the molecular mechanisms found to orchestrate the escape response (*Donnelly et al., 2013*). Our results also hint at a possible answer as to how reorientations of 180° are accomplished: the worm could use its own body as a 'guide' for reorientation. During the omega turn, the distribution of $a_3$ peak amplitudes (*Figure 3D* [Omega turn, inset]) lies close to a value of 15: the lowest $a_3$ value that generates a self-touching body shape. This suggests that the worm might have evolved to coil until it just intersects its own body, which it then slides along to find its way back.

While the omega turn has previously been considered as a single class of *C. elegans* behavior, our analysis of the amplitudes of the curvature mode $a_3$ pulses associated with deep coils, reveals the presence of distinct subpopulations. In foraging, we show that 'classic' omega turns, featuring

the signature $\Omega$ body shape, primarily reorient the worm to the ventral side, while delta turns reorient the worm dorsally by over-turning through the ventral side. These deep dorsal and ventral reorientations occur independently in time with approximately equal rates, which is important if there is to be no overall bias in the trajectories. On the other hand, in an evoked escape response, we observed only $\Omega$-type turns with reorientations of ~180°.

While distinct in visual appearance, omega and delta turns differ only in the amplitude of the curvature mode, and we have shown that these behaviors are discretely separable during foraging. Interestingly, the neuronal basis for omega bend initiation and execution has been studied in some detail (*Gray et al., 2005*), where in particular the SMD and RIV motor neurons are, respectively, implicated in the amplitude and the ventral bias of the turn. Coiling is also observed in other contexts, including a variety of mutants (*Yemini et al., 2013*; *Nagy et al., 2015*), and we expect that our methods will be useful in further analyzing such shapes, and as a guide for uncovering coiling behavior.

Deep turns and reorientations form an important component of the taxis strategy of *C. elegans* (*Croll, 1976*; *Pierce-Shimomura et al., 1999*; *Gray et al., 2005*; *Stephens et al., 2010*; *Salvador et al., 2014*). Under foraging and chemotaxis conditions, these behaviors are seemingly stochastic (*Srivastava et al., 2009*; *Gallagher et al., 2013*), producing a broad distribution of reorientation angles analogous to tumbling in the bacteria *E. coli* (*Berg and Brown, 1972*). However, unlike bacterial tumbling (which occurs through an instantaneous switch in the rotation direction of a molecular motor and the resulting unbundling of the flagellar tail, see, e.g., *Berg, 2006*) the worm's reorientation is driven by a long, controlled sequence of stereotyped postural changes. Thus, an important question is how the worm effectively randomizes its direction of motion. We have shown here that half the variability in *C. elegans* foraging reorientations is due simply to the initial random choice of delta or omega turns. However, even the level of stochasticity can be modulated, as evidenced by the largely deterministic reorientation in the escape response, differing response variability depending on the strength of a thermal stimulus (*Mohammadi et al., 2013*), and the slow adaptation of the reversal rate (*Gray et al., 2005*; *Stephens et al., 2011*). Overall, such a combination of behaviors, flexible and stochastic combined with patterned and deterministic, is likely to be observed even in more complex organisms, including humans. In initiating the detailed analysis of *C. elegans* turning behavior, we hope that our work offers a first step towards a general understanding of these processes.

## Materials and methods

### Data

We used two datasets encompassing both foraging and escape response behavioral conditions (*Broekmans et al., 2016a*). The foraging data were explored previously (*Stephens et al., 2011*); for more details on data collection, see also (*Stephens et al., 2008*). In short, young L4-stage *C. elegans* N2-strain worms were imaged with a video tracking microscope at $f = 32\,\mathrm{Hz}$. Worms were grown at 20°C under standard conditions (*Sulston and Brenner, 1974*). Before imaging, worms were removed from bacteria-strewn agar plates using a platinum worm pick, and rinsed from *E. coli* by letting them swim for 1 min in NGM buffer. They were then transferred to an assay plate (9-cm Petri dish) that contained a copper ring (5.1 cm inner diameter) pressed into the agar surface, preventing the worm from reaching the side of the plate. Recording started approximately 5 min. after the transfer, and lasted for 2100 s (35 min). In total, data from *N* = 12 worms was recorded. The second dataset, the 'escape response' condition, was recorded following procedures as described in ref. (*Mohammadi et al., 2013*). In short, worm recordings took place in a temperature-controlled room (22.5°C ± 1°C). A 100 ms, 75-mA infrared laser pulse from a diode laser ($\lambda$ = 1440 nm) was administered to the head of the worm, raising the temperature in a FWHM-radius of 220 m by ~0.5°C. 10 s of pre-stimulus behavior and 20 s of post-stimulus behavior were recorded at a frame rate of 20 Hz. Each worm was only assayed once, to prevent adaptation. In total, *N* = 92 worms were recorded, of which *N* = 91 successful trackings were used in the final analysis.

## Image processing and shape reconstruction

All movie frames were converted to binary images and cropped, using standard image processing functions in MATLAB (R2014b, The Mathworks, Natick, MA) (*Stephens et al., 2008*). For faster processing, before analysis with the inverse tracking algorithm, the foraging data was down-sampled to 16 Hz by dropping every second frame. To reconstitute an image of a worm with a body posture $\boldsymbol{p} = (a_1, \ldots, a_5)$, we first calculated the vector of backbone tangent angles from $\boldsymbol{\theta} = \sum_i p_i e_i$, with $e_i$ the $i$'th eigenworm. Knowing the total arc length $l$ of the worm, we could calculate the position of each of the 100 points along the backbone. At each backbone point $j$, we then drew a filled circle with radius $r_j$ to capture the worm's body thickness (see also *Figure 1G,H*) and thus create the worm image. Circle radii $r_j$ for a particular worm were computed from movies of uncrossed worm postures for that specific worm. In each such frame, after finding the centerline (backbone) and outline of the worm (*Stephens et al., 2008*), we could find $r_j$ as the minimum distance between backbone point and outline. This was averaged across all frames. Similarly, the total arc length $l$ of the worm was computed by averaging across frames. For the error function described below, the overall orientation of the worm in the image is important, and we generate images of worms in all possible orientations by adding an overall orientation value $\langle \theta \rangle \in [0, 2\pi)$ to the backbone tangent angle vector. This gives us a full backbone vector $\boldsymbol{\theta}_{\mathrm{F}} = \langle \theta \rangle + \sum_{i=1}^{5} a_i e_i$. For the postural dynamics, the eigenworm shape projections were taken from *Stephens et al. (2008)*. Recomputing the eigenworms on the fully-tracked data here showed only minor changes (see *Figure 2—figure supplement 1*).

## Image error function and inverse algorithm

The shape error function compares two binary worm images $\mathbf{W}_1$ and $\mathbf{W}_2$, and is computed as $f_{\mathrm{err}} = f_{\mathrm{outline}} \cdot f_{\mathrm{pixel}}$. For $f_{\mathrm{outline}}$, we calculate a set of tangent angles $\psi$ to the perimeter of the worm shape (*Figure 2A*, bottom left). We find the 4-connected outline of the worm in the binary image $\mathbf{W}_i$, fit a spline through these points, and discretize it into 201 segments sampled at equal arc length. The 200 resulting angles between the segments form a vector $\boldsymbol{\psi}_i = (\psi_{i,1}, \psi_{i,2}, \ldots, \psi_{i,200})$; the total length of the segments is $\ell_i$. $f_{\mathrm{outline}}$ is now $f_{\mathrm{outline}} = C_0 |\boldsymbol{\psi}_1 - \boldsymbol{\psi}_2|^2 + C_1 (\ell_1 - \ell_2)^2$, for arbitrary constants $C_0$ and $C_1$. Note that the value of $f_{\mathrm{outline}}$ is sensitive to the choice of starting points for tracing the 4-connected outline in each image; this is resolved by choosing the pair of starting points that minimizes $f_{\mathrm{outline}}$. For $f_{\mathrm{pixel}}$, we first align the images $\mathbf{W}_1$ and $\mathbf{W}_2$ so that their centroids overlap. Each image is then divided into a grid of 10x10-pixel 'blocks' (*Figure 2A*, bottom right). For each block $(j, k)$ $(j = 1, \ldots, n;\ )$ in image $\mathbf{W}_i$, the fraction $d_i(j, k)$ of black pixels in the block is calculated. This coarse-graining into blocks allows for, e.g., minor inaccuracies in the generation of worm images from mode values, without affecting the error function. We then calculate $f_{\mathrm{pixel}}$ as $f_{\mathrm{pixel}} = \frac{1}{nm} \sum_{j,k} (d_1(j, k) - d_2(j, k))^2$. In earlier trials, we found that using five postural eigenmodes gave us significantly better tracking results than only using four. Since our error function is sensitive to the overall rotation of the worm, we amended the five-dimensional posture space with an extra dimension for the overall orientation $\langle \theta \rangle$. This means that the search space for our algorithm is six-dimensional, with 5 postural dimensions, and 1 rotational dimension. To find a tracking solution for a frame, we ran hundreds of pattern searches (using MATLAB's 'patternsearch' function) from randomly distributed starting points in search space, with the error function described above as objective function. Only solutions with an error value less than 1.0, a threshold value obtained through trial-and-error, were kept. Solutions within a given hypercube of dimensions $[3.0, 3.0, 3.0, 3.0, 2.5]$ were merged, leaving only the solution with the lowest error value. This finally resulted in zero, one, or more potential tracking solutions per movie frame. To speed up the optimization, we applied two additional constraints. Firstly, we bounded the absolute value of the eigenmodes to $(18, 18, 34, 12, 6)$, for each of the five modes respectively. We verified that the distributions of eigenvalues $a_i$ found in our tracking data tailed off before reaching these limits. Secondly, we set a limit to the maximum local curvature of the worm's backbone, so that elements in the resulting theta vector that are 10 indices apart must not be different by more than 1.95 rad. This limit rules out body shapes that were unnaturally coiled.

Importantly, we note that our inverse problem is fundamentally ill-posed: multiple body postures may produce the same two-dimensional worm image (e.g., *Figure 2B*, bottom) and for each movie frame $j = 1, \ldots, N$, we generally find multiple potential solutions which we label $\{\boldsymbol{p}_j^k\}$, with

$k = 1, \ldots, M_j$. Even for simple, non-crossed postures, there can be two solutions ($M_j = 2$), corresponding to the swapped locations of the head and tail. Across the movie, we label the indices of the correct solutions as a vector $\boldsymbol{b} = (b_1, \ldots, b_N)$. We explicitly allow $b_j = 0$ in case the optimization process fails, and use a cubic spline to interpolate across any such gaps. Let us call the point in posture space for movie frame $j$, resulting from this interpolation step, $\tilde{\boldsymbol{p}}_j(\boldsymbol{b})$. To find $\boldsymbol{b}^*$ for the full, correct tracking solution of the movie, we seek the solution vector that minimizes the total sequence error $E(\boldsymbol{b}) = \sum_{j=1}^{N} f_{\text{err}}[\mathbf{W}_j, \tilde{\mathbf{W}}(\tilde{\boldsymbol{p}}_j(\boldsymbol{b}))]$. We constrain the mode changes between two successive frames to be below $v_{\text{max}}$, which simply reflects the fact that the worm can only change posture continuously.

## Tracking pipeline

In a first pass of the data, the 'classic' worm tracking algorithm based on image thinning was used on all frames (*Stephens et al., 2008*). This fast algorithm yields high-accuracy tracking results for frames with simple, non–self-overlapping body shapes. It also automatically labels crossed frames. For the foraging dataset, the data were cut into smaller segments to allow for faster parallel processing. Each segment consisted of a series of non-crossed frames, followed by a series of crossed frames, followed by more non-crossed frames. This effectively segmented the data by deep turns (936 segments in total for the 12 worm trajectories). For the escape response dataset, such segmentation was not necessary, due to the smaller size of the data for each worm. Frames that were labeled by the 'classic' algorithm as 'crossed' were tracked using the inverse algorithm described above. The result was an interpolated, smooth trajectory through posture space. When using this pipeline as-is, the algorithm would occasionally swap the locations of head and tail between frames. To resolve head/tail orientation correctly throughout a segment, we implemented four steps. (1) During the filtering and interpolation step, we allowed the algorithm to pick, for each non-crossed frame, not just the solution given by the 'classic' algorithm; it could also pick an alternative version in which head and tail were swapped (this version can be trivially computed). (2) We explicitly included a limit for the maximum change of overall orientation $\langle \theta \rangle$ between frames of $\sim \pi$ rad per second in the maximum velocity vector $v_{\text{max}}$. Any head/tail swaps between frames violate such a maximum change of $\langle \theta \rangle$. (3) After the filtering and interpolation step had produced a full tracking solution, we computed the error for both that tracking solution, as well as a version in which the head and tail were swapped for all frames in the segment. This fixed the overall head/tail orientation for the full segment. (4) As a final check, we manually verified and, if necessary, corrected head/tail orientations during post-processing. A minimal working set of our tracking code, plus a sample movie that can be successfully tracked using the code's default parameters is available on Figshare as detailed in the author response (code: https://figshare.com/s/3ac08fbfec9ae3d5a531, movie: https://figshare.com/s/658dd86e3847d5926257). A minimal working set of our tracking code, plus a sample movie that can be successfully tracked using the code's default parameters is available on Figshare (*Broekmans et al., 2016b*; *Broekmans et al., 2016c*).

## Tracking quality

In total, 92 escape responses and 936 free-crawling segments (each containing one self-overlapping turn; see above) were analyzed. The escape response tracking results were inspected manually, and 91 trackings (99%) were considered successful, as they were visually close to the appearance of the original worm. For the free crawling dataset, instead, after inspection of a representative sample of 236 segments across multiple worms, 96% were estimated to be successful. First, we assessed the quality of our tracking algorithm for non-crossed worm shapes (*Figure 2E*). We used both the 'classic' algorithm and the 'inverse' algorithm to track $N = 15433$ non-crossed frames from the foraging dataset. For each frame, we calculated the Euclidean distance between the two resulting $\boldsymbol{\theta}$ vectors giving the 'inv. tracking' distribution. In the same figure, the 'dim. reduct.' distribution was calculated from Euclidean distances between the full $\boldsymbol{\theta}$ vector from the classic algorithm, and $\boldsymbol{\theta}_{\text{reduct}} = \sum_{i=1}^{5} a_i e_i$, where $e_i$ are the eigenworms from (*Stephens et al., 2008*) (see also *Figure 2—figure supplement 1*). This represents the information lost in only using the first five postural eigenmodes. The 'time res.' distribution represents the Euclidean distance between $\boldsymbol{\theta}$ vectors from consecutive frames in a movie. In *Figure 2F*, we additionally collected a dataset of four movies, featuring visually

distinct types of omega turns. For the $N = 348$ crossed frames in these four movies, backbones were hand-drawn on the worm images, independently from the tracking results. We compared these backbones to the final results of our inverse tracking / filtering and interpolation algorithms. The resulting mode errors $\delta a_i$ are plotted as the blue/dark distributions. We also include the mode errors for the set of 15433 non-crossed frames (yellow).

## Definition of large-amplitude turns

For the escape response data, the largest peak in $a_3$ between $t = 10$ s (the time of the stimulus) and $t = 29$ s was identified as the apex of the omega turn. To locate the end of the omega turn, the first zero of $a_4$ after the apex was found; any point after that root that had $a_3 < 3$ was considered to be the end of the omega turn. This ensured that the negative peak in $a_4$, representing a high-curvature state of the tail at the end of the omega turn, had finished, and that the worm had reached a relatively 'straight' shape. For such straight shapes, the overall orientation $\langle \theta \rangle$ has a straightforward, intuitive interpretation. The same criterion was used, in the opposite direction, to find the start of the omega turn. If no starting point and/or end point of the omega turn could be found, the recording was excluded from the analysis. (In the escape response dataset, this was the case for 15 out of 91 recordings). We used the same criterion to find both omega and delta turns in the foraging condition. For detection of local extrema in $a_3$, a standard peak-finding algorithm was used to detect both minima and maxima (based on the MATLAB 'findpeaks' function, which defines a peak as a data point with a greater value than its immediate neighbors). Only extrema with a minimum prominence of 0.5 were kept, resulting in 1187 large-amplitude $|a_3| \geq 10$ peaks throughout the entire foraging dataset. Some $a_3$ peaks featured smaller sub-peaks in their shoulders; such sub-peaks were discarded.

## Orientation

Orientation changes were computed by comparing the overall orientation $\langle \theta \rangle$ between two reference points around each omega or delta turn. The apex of each deep turn was the largest $a_3$ peak identified previously. The first reference point was the last frame before the turn's apex that featured a 'straight' body shape — i.e., a body shape with a low maximum local curvature. Only for such relatively 'flat' worm shapes does the overall orientation $\langle \theta \rangle$ correspond directly to the intuitive orientation assigned to the worm. Similarly, the second reference point was the first frame after the turn's apex with such a straight body shape. Importantly, our postural tracking algorithm allows us to continuously follow the orientation angle through coiled shapes and this is important for identifying the 'overturning' reorientation effects of delta turns. For the analysis of the worm's reorientation during the escape response (*Figure 3D,E*), $N = 91$ escape responses were analyzed. Each 30-second recording was segmented by first finding the omega turn. After identification of the omega turn, the reversal phase was simply defined as the first frame after the stimulus with a negative body wave phase velocity $d\varphi/dt$, up until the start of the omega turn. The 'post-omega' phase was any data after the end of the omega turn until the end of the recording at $t = 30$ s. For reorientation during foraging, we analyzed the angle change for segments with self-overlapping turns.

## Mutual information between omega and delta-turn event time series

To calculate the mutual information between the omega and delta turns during foraging, we created a binary event time series by first identifying the time of the $a_3$ peak and then binning these times into bins of width 2, 4, 10, or 20 s. We then calculated the mutual information between these binary time series as in ref. (*Strong et al., 1998*). The mutual information was calculated for different relative shifts, ranging from $-60$ to $+60$ s and the results are shown in *Figure 4—figure supplement 2*. Mutual information across time shifts never exceeded ~3% of the maximum entropy of each time series, indicating that these turns occur independently. There is also no apparent spatial correlation (see *Figure 4—figure supplement 3*).

## Omega and delta turn frequency adaptation

In *Figure 4D*, we show how the average turn frequencies for omega and delta turns change over the course of the 35 min foraging experiments. Turns were detected by using the peak detection algorithm outlined above, applying the amplitude boundaries $|a_3| \geq 10$. The total of these extrema

consists of three populations: the tail of a dorsal/ventral symmetric distribution of shallower turns, and two types of ventral deep turns, the delta and omega turns. To find the number of omega turns, we counted the number of $a_3$ peaks with an amplitude between $-20$ and $-10$ in each time window, and subtracted this from the total number of $a_3$ peaks with an amplitude between $+10$ and $+20$. We then computed the average number of omega turns per unit time, across the 12 experiments, in a 10-minute sliding window, shifted across the data in 5-minute steps. The first 200 s of each experiment were discarded. An identical procedure with $|a_3|>20$ gives the number of delta turns. Over the foraging time analyzed in *Figure 4D*, we find $274 \pm 64$ omega turns and $305 \pm 35$ delta turns, where the errors denote bootstrap errors produced by resampling the $N = 12$ different worm recordings with replacement. The equality of turn counts, within error bars, signals an approximate overall balance in turn events, in agreement with the rate calculations. The total turn rate in *Figure 4D* is comparable to previous work (e.g., *Gray et al., 2005*), though there are notable differences in turn definitions and experimental conditions. We also note, however, that there are spatiotemporal fluctuations in the turn counts, with an increased number of both turns, as well as a specific bias towards omega turns near the location of the copper ring, likely reflecting an increased rate of ring-induced escape responses. In addition, we find an early-time bias towards delta turns, during which we believe that the behavior is strongly influenced by the mechanical perturbation of picking. In future work, it will be fruitful to examine these spatiotemporal patterns in a larger experimental arena and with increased turn statistics.

## Acknowledgements

We thank SURFsara (www.surfsara.nl) for help with the Lisa Compute Cluster. ODB was supported by start-up funds from the Department of Physics and Astronomy, Vrije Universiteit Amsterdam. GJS acknowledges funding from the Department of Physics and Astronomy, Vrije Universiteit and The Okinawa Institute of Science and Technology Graduate University. WSR and JBR thank The National Science and Engineering Council of Canada (NSERC).

## Additional information

### Funding

| Funder | Grant reference number | Author |
| --- | --- | --- |
| Natural Sciences and Engineering Research Council of Canada | Discovery Grant | William S Ryu |
| Vrije Universiteit Amsterdam | Startup Funds | Greg J Stephens |
| Okinawa Institute of Science and Technology Graduate University | Unit Funds | Greg J Stephens |

The funders had no role in study design, data collection and interpretation, or the decision to submit the work for publication.

### Author contributions

ODB, GJS, Conception and design, Analysis and interpretation of data, Drafting or revising the article; JBR, Conception and design, Acquisition of data, Drafting or revising the article; WSR, Conception and design, Acquisition of data, Analysis and interpretation of data, Drafting or revising the article

### Author ORCIDs

Onno D Broekmans, http://orcid.org/0000-0001-8849-7100
William S Ryu, http://orcid.org/0000-0002-0350-7507
Greg J Stephens, http://orcid.org/0000-0003-3135-3514

## Additional files

### Major datasets

The following datasets were generated:

| Author(s) | Year | Dataset title | Dataset URL | Database, license, and accessibility information |
|---|---|---|---|---|
| Rodgers JB, Ryu WS | 2010 | Foraging dataset | http://dx.doi.org/10.5061/dryad.t0m6p | Available at Dryad Digital Repository under a CC0 Public Domain Dedication |
| Ryu WS | 2015 | Escape response dataset | http://dx.doi.org/10.5061/dryad.t0m6p | Available at Dryad Digital Repository under a CC0 Public Domain Dedication |

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
