## [Decision Letter]

Thank you for submitting your article "Resolving coiled shapes reveals new reorientation behaviors in *C. elegans*" for consideration by *eLife*. Your article has been reviewed by three peer reviewers, one of whom, Roland Calabrese, is a member of our Board of Reviewing Editors, and another is Gordon Berman (Reviewer #2), and the evaluation has been overseen by Naama Barkai as the Senior Editor.

The reviewers have discussed the reviews with one another and the Reviewing Editor has drafted this decision to help you prepare a revised submission.

Summary:

In this innovative short report, the authors use a computational approach in combination with video recording of movement to study turning in *C. elegans*, both escape turns and stochastic foraging turns. They develop an automated method for disambiguating coiled postures that builds on their previously published postural analyses that has given rise to the postural eigenmodes by PCA; there are 4 principal eigenmodes. They use this technique to describe turns and the locomotion proceeding and following as movement through the eigenmode space, showing how turns involve the third and fourth eigenmodes and basic crawling the first two eigenmodes. They then apply this techniques to show that escape turns are very stereotyped, while foraging turns have two basic forms; the conventionally recognized omega-turn (similar to the escape turn) and the newly described delta-turn corresponding to the serpentine analog of a left-right step. Moreover in foraging these turns are independent at about the same frequency and show parallel declines with foraging duration. All in all this was an enjoyable paper to read. It introduces a new technique that should be of great interest to the worm community and can serve as an inspiration for automated tracking and dimensionality reduction in other systems. Moreover, it makes a significant discovery the delta-turn that makes unbiased foraging possible.

Essential revisions:

1) The authors should emphasize more clearly in the Abstract and Discussion that the laser stimulus only elicits omega-turns.

2) There is concern that foraging in a copper ring where a nearly equal frequency of omega-turns and delta-turns were observed may not be indicative of foraging in an open petri plate, because copper serves as an aversive stimulus. Are the worms actively avoiding the copper boundary and is this influencing the performance of delta-turns?

3) The authors should clarify the developmental stage of the worms tracked in the copper ring experiments. Larval worms are more flexible than adults and this may influence the presence of delta-turns.

4.)The authors should more clearly acknowledge the early work of Croll (1976) and mention that that work reported that worms reorient randomly to dorsal or ventral side when bumping into a bead, and that their new observation may explain how.

5) The authors might do well to mention in their Discussion the neuronal basis for omega bend initiation and execution by the Bargmann lab (Gray et al., 2004, PNAS). They reported that the SMD and RIV motorneurons appear to stimulate omega bends, and that omega bend amplitude is encoded by SMD, and RIV underlies the ventral bias of omega bends. Indeed, dorsal reorienting turns were shown to occur with equal frequency as ventrally oriented omega bends (Figure 5 H) when the RIV motorneurons were ablated. These dorsal omega bends made by ablated worms in Gray et al. may relate to delta turns in intact worms in the present work.

6) We would particularly like to know how the observed eigenmodes change when including the new data as mentioned explicitly in the minor comments. It also would be interesting to determine if the fourth eigenmode shows similar dynamics to the third, as it temporally leads the 3rd (at least in the example shown in Figure 3). Is this earlier signal (likely a head deflection) predictive of whether the worm goes into an omega or a delta turn?

7) We would like the authors to provide code for their analysis in some form (even just sample code, not necessarily beautifully-written software) so that others can more easily evaluate and use their novel methods for tracking, as mentioned in the minor comments).

---

## [Author Response]

*Essential revisions:*

*1) The authors should emphasize more clearly in the Abstract and Discussion that the laser stimulus only elicits omega-turns.*

We have modified the Abstract and the Discussion (third paragraph) to clarify that the laser stimulus only elects Omega-type turns.

*2) There is concern that foraging in a copper ring where a nearly equal frequency of omega-turns and delta-turns were observed may not be indicative of foraging in an open petri plate, because copper serves as an aversive stimulus. Are the worms actively avoiding the copper boundary and is this influencing the performance of delta-turns?*

Over the foraging time shown in Figure 4 we find (274 +/- 64) omega turns and (305 +/- 35) delta turns, where the errors denote bootstrap errors resampled over the *N*=12 different worm recordings. The agreement in counts, within error bars, signals an approximate balance in overall turning rate, in agreement with the rate calculations. The total turn rate in Figure 4 is reasonably consistent with previous work (e.g., Gray et al., 2005), though there are notable differences in turn definitions and experimental settings. We also note, however, that there are spatiotemporal fluctuations in the turn counts. Indeed, as the reviewers suggest, we do see an increased number of *both* types of turns, as well as a specific bias towards omega turns, near the location of the copper ring. This likely reflects an increased rate of ring-induced escape responses. There is also an early-time (approximately the first 200s) bias towards delta turns, which we believe is due to the mechanical perturbation of picking. We have added these details to the Methods section (subsection “Omega and delta turn frequency adaptation”). In future work, we intend to examine these spatiotemporal patterns in a larger experimental arena and with higher turn statistics. We have also made minor changes to Figure 4, correcting both a mislabeling of the color legend as well as an analysis mistake which resulted in slightly higher reported rates in the previous version of our figure.

*3) The authors should clarify the developmental stage of the worms tracked in the copper ring experiments. Larval worms are more flexible than adults and this may influence the presence of delta-turns.*

We have added a sentence to the Methods section (subsection “Data”), stating the young L4 developmental stage of our tracked worms.

*4.)The authors should more clearly acknowledge the early work of Croll (1976) and mention that that work reported that worms reorient randomly to dorsal or ventral side when bumping into a bead, and that their new observation may explain how.*

We have added a reference to Croll (1975) in the Introduction (third paragraph) and to Croll (1976) in the Discussion of the use of deep turns in a *taxis* strategy (fifth paragraph).

5) The authors might do well to mention in their Discussion the neuronal basis for omega bend initiation and execution by the Bargmann lab (Gray et al., 2004, PNAS). They reported that the SMD and RIV motorneurons appear to stimulate omega bends, and that omega bend amplitude is encoded by SMD, and RIV underlies the ventral bias of omega bends. Indeed, dorsal reorienting turns were shown to occur with equal frequency as ventrally oriented omega bends (Figure 5 H) when the RIV motorneurons were ablated. These dorsal omega bends made by ablated worms in Gray et al. may relate to delta turns in intact worms in the present work.

We have added these ideas to the Discussion, in the fourth paragraph.

6) We would particularly like to know how the observed eigenmodes change when including the new data as mentioned explicitly in the minor comments. It also would be interesting to determine if the fourth eigenmode shows similar dynamics to the third, as it temporally leads the 3rd (at least in the example shown in Figure 3). Is this earlier signal (likely a head deflection) predictive of whether the worm goes into an omega or a delta turn?

We have added Figure 2—figure supplement 1, in which we show the original eigenworms (Stephens et al., 2008); those derived from the current foraging data, but without including crossings; and those derived from the fully-tracked foraging data, including crossed frames. While the eigenworm shapes are largely similar, and in all cases 4 modes capture over 95% of the postural variance, the third (turning) eigenmode individually accounts for more variance in the fully-tracked data, as expected given its primary role in describing deep turns.

The comment about the possible predictive value of the fourth mode is an interesting one. We have included in our response (response-fig_a3-vs-a4.pdf) a scatter plot of the peak amplitude of the a4 peak immediately preceding the deep turn, vs. the peak amplitude of the a3 peak itself. The points are color-coded, so that red corresponds to omega turns, and blue to delta turns, as classified in Figure 4. There is a visible link between the two quantities, and a significant overall correlation of 0.73 +/- 0.03. This suggests that the specific turn-type is determined relatively early in the dynamics. We did not include this plot as supplementary material, as it takes us farther afield from the aim of our current manuscript. We do intend to revisit these points in future work clarifying the turning mechanism.

*7) We would like the authors to provide code for their analysis in some form (even just sample code, not necessarily beautifully-written software) so that others can more easily evaluate and use their novel methods for tracking, as mentioned in the minor comments).*

We share your stance on the importance of sharing scientific code, and have uploaded a minimal working set of our tracking code, plus a sample movie that can be successfully tracked using the code’s default parameters, to Figshare: https://figshare.com/s/3ac08fbfec9ae3d5a531

Sample movie: https://figshare.com/s/658dd86e3847d5926257

The datasets are currently marked as private, but anyone with the above links can access the files. We will open up access to everyone as soon as the paper is public.